Assessing availability of European plant protection product data: an example evaluating basic area treated

López-Ballesteros Ana 1 2
Delaney Aoife 2 3
Quirke James 4
Stout Jane C. 5
Saunders Matthew 5
Carolan James C. 6
White Blánaid 7
Stanley Dara A. dara.stanley@ucd.ie 2
1 Department of Agricultural and Forest Systems and the Environment, Agrifood Research and Technology Centre of Aragon (CITA) , Zaragoza , Spain
2 School of Agriculture and Food Science, University College Dublin , Dublin , Ireland
3 National Parks and Wildlife Service, Department of Arts, Heritage and the Gaeltacht , Dublin , Ireland
4 Department of Agriculture, Food and the Marine , Backweston , Kildare , Ireland
5 Department of Botany, School of Natural Sciences, Trinity College Dublin , Dublin , Ireland
6 Department of Biology, Maynooth University , Maynooth , Kildare , Ireland
7 School of Chemical Sciences, DCU Water Institute, Dublin City University , Dublin , Ireland
Anderson Todd
Electronic publication date: 2022 Jul 13
Publication date: 2022
Volume: 10
Electronic Location ID: e13586
Received 2022 Jan 21; Accepted 2022 May 23
Copyright: ©2022 López-Ballesteros et al.
Copyright year: 2022
Copyright holder: López-Ballesteros et al.
License: This is an open access article distributed under the terms of the Creative Commons Attribution License, which permits unrestricted use, distribution, reproduction and adaptation in any medium and for any purpose provided that it is properly attributed. For attribution, the original author(s), title, publication source (PeerJ) and either DOI or URL of the article must be cited.
License URL: https://creativecommons.org/licenses/by/4.0/

Keywords: Pesticides, Exposure, European policy, Agrichemical, Plant protection product, Active ingredients

Funding: Irish Department of Agriculture, Food and the Marine Award 17/S/232 Juan de la Cierva Postdoctoral Contracts FJC2018-038192-I IJC2020-045630-I European Union NextGenerationEU/PRTR MCIN/AEI/ 10.13039/501100011033 MdM-2017-0714 MCIN/AEI/10.13039/50110001103 This work was funded by an Irish Department of Agriculture, Food and the Marine award (PROTECTS Project Ref: 17/S/232). Ana López-Ballesteros was also supported by Juan de la Cierva postdoctoral contracts FJC2018-038192-I and IJC2020-045630-I, funded by MCIN/AEI/ 10.13039/501100011033 and European Union NextGenerationEU/PRTR, and supported by ref. MdM-2017-0714 funded by MCIN/AEI/10.13039/50110001103. James Quirke works for the same agency that funded this research, however, he works in pesticide statistics and is not associated with funding allocation.

==============================
Besides the benefits of plant protection products (PPPs) for agricultural production, there is an increasing acknowledgement of the associated potential environmental risks. Here, we examine the feasibility of summarizing the extent of PPP usage at the country level, using Ireland as a case study, as well as at the European level. We used the area over which PPPs are applied (basic area) as an example variable that is relevant to initially assess the geographic extent of environmental risk. In Irish agricultural systems, which are primarily grass-based, herbicides fluroxypyr and glyphosate are the most widely applied active substances (ASs) in terms of basic area, followed by the fungicides chlorothalonil and prothioconazole that are closely associated with arable crops. Although all EU countries are subject to Regulation (EC) No 1185/2009, which sets the obligation of PPP usage data reporting at the national level, we only found usable data that met our criteria for Estonia, Germany, Finland, and Spain (4 of 30 countries reviewed). Overall, the most widely applied fungicide and herbicide in terms of basic area were prothioconazole (20%, 7% and 5% of national cultivated areas of Germany, Estonia and Ireland) and glyphosate (11%, 8% and 5% of national cultivated areas of Spain, Estonia and Ireland) respectively, although evaluations using application frequency may result in the observation of different trends. Several recommendations are proposed to tackle current data gaps and deficiencies in accessibility and usability of pesticide usage data across the EU in order to better inform environmental risk assessment and promote evidence-based policymaking.

Introduction

The risk of crop loss in modern agricultural production has been minimised over the past century through the use of plant protection products (PPPs) (Kim, Kabir & Jahan, 2017). PPPs are synthetic or natural chemical products intended for preventing, destroying or controlling any pest causing harm to, or otherwise interfering with, the production, processing, storage, transport or marketing of plant-based food and agricultural commodities. Active substances (ASs) contained in PPPs can include, among others, herbicides, fungicides, insecticides, acaricides, nematicides, molluscicides, or plant growth regulators, individually or in combination (FAO, 2006). Global PPP use (in terms of kg applied per hectare) rose steadily during the second half of the 20th century until the beginning of 21th century (Sharma et al., 2019; Zhang, 2018), and in 2019, worldwide PPP use was estimated to be four million tonnes (FAO, 2019). Globally, the quantity (tonnes of AS) of herbicides used is twice that of fungicides and almost four times that of insecticides (Zhang, 2018) while in the EU the quantity of fungicides and herbicides used are similar and both more than double that of insecticides (FAO, 2019). Factors influencing usage trends of PPPs include pest pressure, pest-control effectiveness, accessibility, cost and regulatory status (Barzman et al., 2015). In 2014, amide-based compounds, including phenoxy hormone products and bipiridils led global herbicide usage by weight, whereas inorganic dithiocarbamate and triazole compounds, and organophosphates, pyrethroids and carbamates, account for the greatest weight of fungicides and insecticides applied respectively (Zhang, 2018).

While the use of PPPs has contributed to agricultural production and food security (Cooper & Dobson, 2007), research has revealed potentially harmful effects of reliance on these substances. Drawbacks include issues such as human health impacts (Tanner et al., 2011; Alavanja & Bonner, 2012; Anderson & Meade, 2014), development of pest resistance, resurgence and secondary pest outbreaks (Evenson & Gollin, 2003; Oerke, 2006) and environmental contamination. The quantity of PPPs present in the environment is related to the geographic extent of application, the amount applied, and also to the persistence of their associated ASs together with their metabolites. Residues of PPPs are widespread in soil where crops are grown, and 39% of residues found in the EU were considered persistent or very persistent (Silva et al., 2019), with the most common residues reported being glyphosate, AMPA (a metabolite of glyphosate) and DDE (dichlorodiphenyldichloroethylene), a derivative of DDT, which has not been licensed for use in the EU since 1983 (European Commission, 2003). The presence of PPPs and their residues in air, soil, water and food can harm non-target organisms, both in the area where they are applied and also the wider landscape. Non-target organisms may come into contact with PPPs through direct application (Boutin et al., 2014), inadvertent contamination including drift (Morrissey et al., 2015), through their diet (de Snoo & Luttik, 2004; Zioga et al., 2020), or they may be affected by a reduction in food availability (Eng, Stutchbury & Morrissey, 2017; Hallmann et al., 2014). In addition, coformulants, that are combined with ASs in PPP formulations can have synergistic or antagonistic effects on AS toxicity (Takács et al., 2017) or be toxic in themselves (Mesnage, Benbrook & Antoniou, 2019). Where they occur, such impacts are taxon-specific and may affect species that are different to the target organism. Some bees, for example, are significantly vulnerable to some fungicides and insecticides (Arena & Sgolastra, 2014; Bernauer, Gaines-Day & Steffan, 2015; Main et al., 2018) and also to the synergistic impacts of both (Sgolastra et al., 2017). Because of the species-specific responses to different ASs, broad investigations of the impacts of PPPs on organisms can yield complex results, as illustrated by the review of Puglisi (2012) on the response of microbial organisms to PPPs which found that herbicides, fungicides and insecticides stimulated terrestrial microbial biomass in some studies and suppressed it in others. However, there is a need to understand what pesticides are used, and how their use is distributed geographically, in order to fully assess risks to non-target organisms and the environment more generally.

Concerns regarding the potential for unintended consequences of PPP use led to the establishment of international PPP usage databases. At a global level, the FAO pesticide use database (FAO, 2019) presents the quantity in tonnes of the major pesticide groups (e.g., fungicides, insecticides, herbicides) used in or sold to the agricultural sector, and their constituent chemical class (e.g., carbamates, organophosphates) while the FAO pesticide indicators database (FAO, 2018a), which in turn relies on the FAO land use database (FAO, 2018b), reports PPP usage in kg/ha of cropland for most countries in the world from 1990 to 2016 without specifying major pesticide groups or ASs. At an EU level, the EUROSTAT database (EUROSTAT, 2019a, EUROSTAT, 2019b) also collates data on PPP sales (kg of PPP major groups) in an accessible format, for EU Member States and another ten European countries (under Regulation (EC) No 1185/2009). Although these databases can be used to assess broad trends in the quantity of PPP sold globally and within the EU (e.g., Zhang, 2018), they do not link pesticide use to specific crop types or indicate the area or geographic extent of pesticide application, both of which are important indicators of environmental risk. Thus, given the absence of real georeferenced data, modelling approaches have been implemented to try and fill this data gap at a global level. For instance, PEST-CHEMGRIDS models and predicts future spatially-explicit pesticide use of some key ASs based on FAO and USGS data (Maggi et al., 2019).

Some of the most detailed pesticide usage data available globally is collected in the EU (but also see Mesnage et al., 2021). The Sustainable Use of Pesticides Directive (2009/128/CE; hereinafter SUD) established a framework for reducing risks to and impacts upon human health and the environment arising from pesticide use. Regulation (EC) No 1185/2009 was enacted to ensure that detailed, up-to-date and consistent information from Member States would be available for risk assessment and to monitor progress towards the goals of the SUD. Under this regulation, Member States are required to report the area treated with PPPs (ha) and the quantity applied (kg) for major crop types in their jurisdictions. The collection of statistics on PPP usage on the ground is an integral feature of the SUD, as it provides information required to assess the risks posed by PPP usage. However, no central, publicly-accessible data repository has been established for this information at an EU scale, and difficulties in comparing usage data across different nations have been reported (https://eur-lex.europa.eu/legal-content/EN/TXT/?uri=CELEX%3A52017DC0109). This is important because failure to provide accessible, comparable data across EU Member States inhibits risk assessment both at a Member State and EU-wide scale.

In this study, we aim to assess whether data on PPP usage collected under the SUD can be readily used to (i) estimate total PPP usage in terms of national area of application (basic area), (ii) compare trends in PPP usage in terms of basic area of application among European countries and (iii) identify the most widely used ASs across Europe in terms of basic area of application. We chose to use the unit of basic area as it implicitly includes the spatial dimension of pesticide usage (contrary to mass units) and it can be the first step in understanding the geographic extent of use. We used Ireland as a case study for the estimation of national PPP usage because agriculture is an important land use, accounting for 65% of the national land area (https://www.cso.ie/en/releasesandpublications/ep/p-syi/psyi2018/agri/cl/), and because a nationwide estimate for PPP use in Ireland has not previously been published. By answering these questions, we aim to demonstrate the availability, accessibility and usability of national PPP usage data in Europe. Overall, this study provides insight into the effectiveness of current legislative tools designed to evaluate the success of some SUD objectives.

Materials and Methods

Irish data on usage of plant protection products

The Irish Government Department of Agriculture, Food and the Marine’s (DAFM) pesticide usage reports from 2014–2017 (http://www.pcs.agriculture.gov.ie/sud/pesticidestatistics/) were used as the source of data relating to PPP usage in crops in the Republic of Ireland. These reports present Irish national statistics on PPP usage collected and collated by DAFM in line with EU Regulation (EC) No 1185/2009 in a four-year cycle for each crop type. Estimates of PPP usage provided by DAFM are based on a survey of a sample of farms within each crop type (Table 1). Although the sample size varied among crop types, in each case the sample was selected to be representative of the range of farms within each crop type in Ireland. Fruit crops were divided into top fruit, including apples and other fruit grown on trees, and soft fruit such as strawberries, raspberries and blackcurrants. The vegetable and arable crop types were limited to those intended for human consumption, while vegetables and grain grown as animal fodder were included in the grassland and fodder report.

Farmers were asked to provide details of the PPPs applied, the date of application, the area to which PPPs were applied and the specific crop grown in each field included in the survey in the 12 months prior to harvest. These data were then used to calculate the area of application (i.e., basic area) of ASs and of PPP groups (i.e., fungicides, herbicides, insecticides, plant growth regulators and other) at a national scale, as well as a number of other variables not considered further. Thus, if several applications of an AS or a PPP group occurred within a single cropped area, that cropped area was only included once in the measure of basic area used in our analysis, and duplicates were discarded. Specifically, the analysed data correspond to tables included in the DAFM pesticide usage reports that document the sample area, national cultivated area, and basic areas of application per AS and PPP group (i.e., fungicides, herbicides, insecticides, plant growth regulators and molluscicides; see Tables C, 4, and 10–13 in 2014 Top Fruits Survey Report, Tables C, 4 and 10–18 in Soft Fruits Survey Report, D 4, and 10–28 in 2015 Vegetable Survey Report, Tables C, 3, and 9–19 in 2016 Arable Survey Report, Tables C, 4, 10–20 in 2017 Grassland and fodder Survey Report). In these reports, ASs applied via seed treatments are explicitly reported. The tabulated data was provided upon request by DAFM in csv format. Full details of the methodology used to collect and process the raw data are presented in the pesticide usage reports (http://www.pcs.agriculture.gov.ie/sud/pesticidestatistics/).

Table 1 Irish pesticide usage data collection summary.

Crop type, the year for which data were collected and sample size (number of farms, area surveyed and proportion of each crop type surveyed) for each of the Irish pesticide usage surveys included in this report.

Crop type	Survey year	Number of farms surveyed	Area surveyed (ha)	Area surveyed (% of total crop area)	
Top fruit	2014	23	492	79.7	
Soft fruit	2014	26	187	56.0	
Vegetable	2015	109	2,902	61.2	
Arable	2016	260	23,199	7.6	
Grassland and fodder	2017	530	33,187	0.7	

European data on usage of Plant Protection Products

Sources of national PPP usage data for European countries were sought in September 2019 from the websites of national institutions with competencies in the implementation of the following European legislation: Residues - Regulation (EC) 396/2005, Sustainable Use Directive 2009/128/EU and Pesticide Regulation (EC) No 1107/2009. This information was made publicly available by the European Commission (https://ec.europa.eu/food/sites/food/files/plant/docs/pesticides_legis_national-authorities_en.pdf) for Norway, Iceland and the 28 EU Member States. We selected the most up-to-date national databases that presented basic area of ASs applied to specified crop types in tabulated format. If such data were not presented in tabulated format, they were not considered to be accessible. In the case of Ireland, the analysed tabulated data were provided by DAFM in csv format, and these data coincide with tables included in the Irish PPP usage reports from which data are not directly exportable. Standardisation levels among databases were assessed in terms of units used to report PPP usage and database structure (i.e., levels of classification for crop types and PPP groups). Although data for all countries were collected in various ways, data presented in reports represent estimations or measures of use at a national scale and so are directly comparable.

We assessed whether the accessible databases were comparable in terms of their time-span, reporting units and the categorization scheme used to define PPP groups and crop types. We then compared the basic area of ASs in countries with comparable available data (Estonia, Finland, Germany and Spain) with those of Ireland. We then organised ASs into standardised major PPP groups to facilitate comparison of broader categories among countries. We included any pesticide groups reported with multiple functions to our standardised PPP groups as follows: (i) fungicides include fungicides-bactericides and fungicides-plant growth regulator groups, (ii) herbicides include herbicides-moss control groups, (iii) insecticides include insecticides-acaricides and (iv) the group labelled “other” includes molluscicides and pheromones. In addition, we extracted national cultivated areas (all the land dedicated to agriculture in a country) from the EUROSTAT databases of national utilised agricultural area (code TAG00025) for the years where the most recent PPP usage data was available for each country (i.e., 2015, 2018, 2017, 2017 and 2013 for Estonia, Finland, Germany, Ireland and Spain, respectively). To compare the use of major PPP groups, we utilised maximum (national cultivated area) and minimum (maximum basic area of the most widely used AI) values for each country. Presenting the area of AS application as a range was appropriate since, in many cases, multiple different ASs were applied to the same parcel of land in a reporting period. Thus, adding together all AS basic areas reported resulted in a clear overestimate of the area where PPPs are applied in each country, which in some cases exceeded the total national cultivated area.

Results and Discussion

Irish usage of plant protection products

The area of land under agricultural management in Ireland is dominated by grassland and fodder systems (94%; Table 2), with grasslands accounting for for 99.4% of this land use type. Arable crops are grown on 6% of agricultural land, and vegetable and fruit cultivation each cover less than 0.1%, with organic agriculture occupying less than 2% of total agricultural area. Based on DAFM national statistics, PPPs are applied in more than 90% of the national area of arable, vegetable and fruit crops compared to under 10% of the national area occupied by grassland and fodder (Table 2). Although grassland and fodder systems have the lowest PPP use in relation to area, the basic area of PPP use in grasslands and fodder exceeds that of other crop types, and this reflects the prominence of grasslands in Irish agriculture.

Table 2 Sample, national cultivated and national basic areas where PPPs were applied by crop type for Ireland.

Sample, national cultivated and national basic areas where PPPs were applied (in ha and % of total cultivated area) by crop type for Ireland. Data source: Irish Government Department of Agriculture, Food and the Marine’s Pesticide (DAFM) usage reports.

Crop type	Year sampled	Sample area (ha)	National cultivated area (ha)	National basic area of PPP application (ha)	% Total cultivated area in which PPPs are applied	
Arable	2016	23,199	306,092	305,744	99.89	
Soft & top fruits	2014	679	951	924	97.18	
Vegetables	2015	2,831	4,635	4,314	93.07	
Grassland & fodder	2017	33,187	4,652,044	431,154	9.27	

Herbicides are the most widely used PPPs in Ireland geographically with an estimated national basic area more than twice the basic area of fungicides (Fig. 1A). Fungicide is the second most widely used PPP group followed by insecticide and plant growth regulators, and respectively these represent 44%, 32% and 24% of the area where herbicides are applied. Molluscicide application, at only ∼12,500 ha nationally, is less widespread. Although most PPPs are applied as spray, seed treatments represent a similar area to that treated with fungicides (∼280,000 ha, Fig. 1A).

Figure 1 Basic area of main PPP groups used in Ireland.

Estimated national basic area (103 ha) for the main groups of plant protection products (PPPs) used in Ireland (A) together with the percentage applied to the different crop type areas (B). Soft and top fruit crops are not visible; while PPP usage is high in these systems, they represent a very small geographic area nationally. Data source: Irish Government Department of Agriculture, Food and the Marine’s Pesticide (DAFM) usage reports.

The area over which different PPP groups are utilized is strongly related to the crop type (Fig. 1B). For instance, arable crops have the largest proportional areas where fungicides (92.6%), insecticides (95.5%), molluscicides (74.3%), and plant growth regulators (99.6%) are used. Likewise, PPPs are more widely applied as seed treatments in arable systems (93.9%) although these can also be found in grassland and fodder systems (6.1%). The greatest basic area of herbicide application occurs in grassland and fodder systems followed by arable crops, which represent 58.4% and 41% of total national area of herbicide application, respectively. Although PPPs are applied to 97% of the cultivated area designated to soft and top fruits, the percentage area of PPP application in these systems remain marginal (<1%) compared to the rest of crop types, given the small area they represent in the total Irish agricultural land (Table 2). However, it is likely that intensity of use may be much higher in soft and top fruit systems which suggests the importance of considering both basic area and intensity of use in fully assessing risk.

We found that in arable systems and for cropped areas where application date was recorded (76% of total records), fungicides were sprayed more frequently (mean number of application dates = 4.8, s.e. = 0.12) than herbicides (mean number of application dates 2.8, s.e = 0.05). This disparity in application intensity is likely to explain why we found that herbicides are used over the widest geographic area in Ireland, despite the finding of Zhao et al. (2013) that fungicides are the most intensively utilized PPP group in Ireland when looking at units of mass per area (kg of AS per km2). Nevertheless, the use of herbicide over a large geographic area, even at a low intensity, is not without consequence. In 2020, herbicide was the most commonly found pesticide type in the 33 Irish public water supplies that failed to meet the EU pesticide standard set under the European (Drinking water) Regulations 2014 (Environmental Protection Agency, 2021). Both basic area and intensity (or frequency) are therefore relevant when considering the risks to the environment and human health associated with pesticide usage.

Because the usage intensity of PPPs varies depending on crop type, land-use changes over time are likely to have an impact on the presence of PPPs and their residues in the environment, and on exposure of non-target organisms. In Ireland, arable land is largely concentrated in the south-east of Ireland and a smaller area in the north-west (Environmental Protection Agency, 2018), and we may expect greater use of PPPs in these areas than in grass dominated landscapes. However, some parts of Ireland are characterised by frequent land use change between arable and grassland systems (Zimmermann & Stout, 2016), so the area in which PPPs and their residues occur may exceed the area of application captured in a 12-month SUD reporting period. In addition, a transition from a grassland-dominated system to a more diversified system that includes more arable and horticultural crops in the future (for example to adapt to climate change or diversifying markets) could lead to an increase in the extent of PPP use, if current agricultural practices persist.

Regarding the area (ha) where specific ASs have been applied, the prevalence of some ASs over others can be observed for the main PPP groups (Fig. 2). For instance, the two most widely used ASs geographically, the fungicides chlorothalonil, a non-systemic and broad-spectrum PPP (now banned in the EU since 2019), and prothioconazole, a systemic PPP, were each applied over more than 200,000 ha nationally. These two ASs were most frequently used in arable crops, especially in barley, wheat, oats and winter oilseed rape cultivation; however, they were also utilised in vegetable, and grassland and fodder cropping systems. The most widely used herbicides in Ireland were fluroxypyr, glyphosate and 4-chloro-2-methylphenoxy acetic acid (MCPA) with national basic areas ranging from approximately 129,000 ha to and 271,000 ha. These ASs are commonly applied in arable (barley, wheat and oat cultivations) and grassland and fodder systems, and glyphosate and MCPA were also used in vegetable and fruit crops. The most widely used herbicidal ASs are all systemic PPPs. Fluroxypyr and MCPA are synthetic auxins, while glyphosate is an enzymatic inhibitor of 5-enolpyruvylshikimate-3-phosphate (EPSP) synthase. The use of MCPA in many agricultural systems across Ireland is likely to explain why two-thirds of public water supply samples that failed to meet the pesticides standard in 2020 contained MCPA (Environmental Protection Agency, 2021), but specific chemical characteristics of MCPA, the conditions in which it is applied, and that it is the main compound used for rush control are also likely to play a part. A single AS, lambda-cyhalothrin, dominates insecticide use and was applied to ∼150,000 ha, including arable (barley, wheat, oats, oilseed rape and potatoes), grassland and fodder and vegetable (e.g., carrots, parsnips, cabbages, spinach) crops. Chlormequat, the dominant plant growth regulator, was used across a slightly larger area nationally (∼170,000 ha), mainly in arable crops (barley, wheat and oats).

Figure 2 Basic area of most widely used active substances (ASs) in Ireland.

Estimated national basic area (103 ha) of the most widely used active substances applied in Irish agriculture. Agricultural area (grey) includes grassland and fodder, soft and top fruits, vegetables and arable systems. Data source: Irish Government Department of Agriculture, Food and the Marine’s Pesticide (DAFM) usage reports.

Available data on European usage of plant protection products

While PPP sales data show a high level of harmonisation and are available in a common repository for all EU countries (EUROSTAT, 2019a, EUROSTAT, 2019b), we found that available PPP usage data in terms of area of application is sparse, difficult to access, and different reporting formats makes usability challenging. Only four EU Member States (Estonia, Finland, Germany and Spain) reported PPP usage data that were available, accessible and in a format that enabled a direct assessment and comparison for our work. For the remaining countries, PPP usage data were either not publically available (19 of 30 total), or in a non-accessible and non-usable format due to the reporting units (three countries) and format (e.g., text reports in native languages where data tables were embedded in PDF files; four countries including Ireland). Reporting frequency varies among countries, as is reflected in the different database publication dates (Table 3). Although the mandatory frequency set by the SUD is five years, some countries report the data annually (Germany and Estonia), while others exceeded this five-year time span (Spain) or report different crop types every year (Ireland), which makes a comparison of PPP usage across countries in the same year impossible.

Table 3 Accessed PPP usage public data for each country together with the disaggregation levels utilized to classify PPP groups and crop types.

Accessed PPP usage public data for each country together with the disaggregation levels utilized to classify PPP groups (i.e., disaggregation levels 1 and 2 corresponding to major groups, and active substances, respectively) and crop types (disaggregation levels from 1 to 3 corresponding to broader to more specific crop types). Blank cells mean that no data was available for a given disaggregation level.

Country	Year	PPP groups	Crop types	Area	Uncertainty indicator provided	
		Disaggregation levels	Disaggregation levels			
		1	2	1	2	3			
Estoniaa	2015	4	140	11	8	45	Basic	None	
Finlandb	2018	4	–	19	–	–	Basic	None	
Irelandc	2014–2017	6	215	5	–	56	Basic & treated	None	
Germanyd	2017	5	222	9	–	–	Basic	Confidence interval	
Spaine	2013	6	272	7	–	–	Basic & treated	None	
Notes.

a Data source: https://andmed.stat.ee/en/stat.

b Data source: https://statdb.luke.fi/PXWeb/pxweb/en/LUKE/LUKE__02%20Maatalous__04%20Tuotanto__34%20Kasvinsuojeluaineiden%20kaytto%20maataloudessa/02_Kasvinsuojeluainekaytto.px/.

c Data source: http://www.pcs.agriculture.gov.ie/sud/pesticidestatistics/.

d Data source: https://papa.julius-kuehn.de/index.php?menuid=33.

e Data source: https://www.mapa.gob.es/es/estadistica/temas/estadisticas-agrarias/agricultura/estadisticas-medios-produccion/fitosanitarios.aspx. Accessed in September 2019.

The structure of the available databases also differed in terms of the PPP groups and the reported crop types that accompany usage data. Different disaggregation levels were reported for different countries with Ireland and Estonia having the highest number of disaggregation levels for PPP groups and crop types, respectively. Furthermore, the type of data reported are not consistent among databases. For instance, only Ireland and Spain include molluscicides as a major PPP group, while pheromones were included in the German pesticide usage data. The four PPP groups that are present in all national databases are fungicides, herbicides, insecticides and plant growth regulators, assuring comparability among countries for these major PPP groups. However, for some countries, some of these PPP groups include pesticides with more than one function, such as insecticides-acaricides, fungicides-bactericides, herbicide-moss killers, and fungicides-plant growth regulators. The categorisation of crop-types is more variable than that of PPP groups, and we could not find any crop type that is common for all countries. Among all national databases, there were 11 crop types found in more than one country for all disaggregation levels: potatoes (2 countries), vegetables (3), winter (4) and spring wheat (3), cabbages (2), carrots (3), onions (3), peas (2), and spring (2) and winter barley (3). The two databases with the highest crop-type disaggregation levels, Ireland and Estonia, are also the most similar in their categorisation of crop types.

With regard to the units used for the area of applied PPP, only the Spanish and Irish databases include values of both basic and treated/sprayed areas, which together with the amount of AS applied per unit of area (e.g., in kg/ha) can give us an estimate of PPP use intensity. However, working with intensity data in terms of the weight applied per unit area can be problematic, as although the quantity of the AS applied can be closely related to its toxicity this differs between ASs, so the risk of environmental contamination may not be proportional to the weight of a substance that is applied. Furthermore, the quantity in a single application may be misleading. For example, trends of PPP usage over time from Northern Ireland have shown that while the weight of PPP per application to arable crops has decreased, the number of applications to the same piece of land have increased, resulting in a similar total quantity of PPP applied over time (Jess et al., 2018). Since both measures, area and intensity/frequency of application, carry complementary information, further specifications must be set on how exactly data should be reported using these specified units in order to enable data comparison across EU countries.

Generally, no distinction is made between the application method (seed treatment vs spray) used to apply each AS. Only Ireland provided explicit information on the area where ASs were applied via seed treatments whereas the Finnish database excluded those areas that were exclusively exposed to seed- or seedling-treatment. Uncertainty estimates were only incorporated in the German database, where confidence intervals of basic areas were included (Table 3). Although we successfully compared pesticide usage data from a small number of countries, the observed reporting differences among EU member states made comparisons of PPP usage data at an EU scale difficult, even for countries with publicly available usage data, confirming the findings of the European Commission (2017). This inhibits the possibility of using this information to assess the potential hazards and risks of PPP use for a variety of purposes, including the assessment of potential environmental contamination.

Comparison of plant protection products usage among EU member states

For countries where data were available, agricultural systems differ in terms of area and crop types. For instance, a higher proportion of the national area is dedicated to agriculture in Ireland (65%), Germany (47%) and Spain (46%), compared to Estonia (22%) and Finland (7%). In addition, more than half of the total cultivated area was dedicated to arable crops for all countries except Ireland, where the agricultural system is clearly grassland-dominated (90% of total cultivated area; Table 4).

Table 4 Country and cultivated areas.

Country and cultivated (103 ha) areas, together with the fraction of cultivated area (CA) designated to the distinct crop types. Data source: EUROSTAT TAG00025 and org_cropar databases.

Country	Country area
(103 ha)	National cultivated area
(103 ha)	Organic crops (% CA)a	Arable
(% CA)	Permanent grasslands
(% CA)	Permanent crops
(% CA)	Kitchen crops
(% CA)	
Estonia	4,522.70	993.60	15.68	67.02	31.69	0.33	0.97	
Finland	33,844.00	2,271.90	13.09	98.71	1.06	0.15	0.06	
Germany	35,737.60	16,687.30	6.82	70.30	28.25	1.19	0.01	
Ireland	6,979.70	4,489.21	1.66	10.02	90.53	0.04	0.00	
Spain	50,594.40	23,494.57	6.85	52.40	27.20	19.93	0.47	
Notes.

a This value includes agricultural areas that are fully converted and under conversion to organic farming. This value excludes kitchen gardens.

Almost all of the most widely used ASs in terms of basic area were fungicides and herbicides in each country (Table 5), with the exception of plant growth regulators chlormequat and trinexepac in Ireland and Germany, and the insecticide dimethoate in Spain. A number of ASs occur among the top five most widely applied geographically for several countries. For example, the systemic fungicide prothioconazole is notable for its application to 20% of the national cultivated area (CA) in Germany, a figure equivalent to 9.5% of the total land area of Germany, and is also extensively used in Estonia (7% CA) and Ireland (5% CA). Another widely used systemic fungicide is tebuconazole, which is reported within the top five ASs in terms of basic area of Estonia (12% CA) and Germany (19% CA). The most widespread herbicidal AS reported is glyphosate, which is among the most widely applied ASs in Ireland (>5% CA), Estonia (>8% CA) and Spain (>11% CA). This prevalence of glyphosate, prothioconazole and tebuconazole among the countries assessed is in accordance with results from Silva et al. (2019), who showed that residues of these compounds are commonly found in European soils. Importantly, most of the Silva et al. (2019) study’s soil samples contained residues of multiple ASs, a fact that is particularly relevant for non-target organisms, which may often be exposed to more than one compound at a time potentially leading to combined negative impacts of multiple ASs, a phenomenon commonly known as the “cocktail effect” (Relyea, 2009; Rivera-Becerril et al., 2017; Soil Association, Pesticide Action Network UK, 2019). Unfortunately, this trend cannot be investigated in PPP-use databases currently available as there is no spatially explicit or georeferenced information.

Table 5 National basic areas (ha) treated with the top five most widely used active substances per country.

National basic areas (ha) treated with the top five most widely used active substances in Estonia, Germany, Ireland and Spain, together with the proportion these represent of the total national cultivated area (CA) and country areas. Data sources: publically accessible data of national plant protection products use, and EUROSTAT TAG00025 and org_cropar databases.

Country	PPP group	Active ingredient	Basic area (ha)	% CA	% country area	
Estonia	Fungicides	Tebuconazole	114,779.02	11.55	2.54	
Herbicides	Glyphosate	83,335.97	8.39	1.84	
Herbicides	Florasulam	79,537.10	8.00	1.76	
Fungicides	Prothioconazole	73,079.43	7.36	1.62	
Herbicides	Iodosulfuron-methyl-sodium	70,017.82	7.05	1.55	
Germany	Fungicides	Prothioconazole	3,383,028.00	20.27	9.47	
Fungicides	Tebuconazole	3,251,387.00	19.48	9.10	
Plant Growth Regulators	Trinexapac	3,033,290.00	18.18	8.49	
Fungicides	Epoxiconazole	2,963,429.00	17.76	8.29	
Herbicides	Flufenacet	2,585,973.00	15.50	7.24	
Ireland	Herbicides	Fluroxypyr	271,137.11	6.04	3.88	
Fungicides	Chlorothalonil	239,146.51	5.33	3.43	
Fungicides	Prothioconazole	219,424.13	5.04	3.14	
Herbicides	Glyphosate	218,858.60	4.84	3.14	
Plant Growth Regulators	Chlormequat	163,052.85	3.72	2.34	
Spain	Herbicides	Glyphosate	2,588,693.10	11.02	5.12	
Herbicides	2,4-D acid	1,550,798.90	6.60	3.07	
Herbicides	Tribenuron-methyl	1,419,758.40	6.04	2.81	
Fungicides	Copper oxychloride	1,287,566.80	5.48	2.54	
Insecticides	Dimethoate	826,432.00	3.52	1.63	

The majority of the most widely used ASs for all countries are currently approved for use with some exceptions. For example, while Irish results showed that both chlorothalonil and clothianidin were widely applied during the last reporting period, and neither of these compounds are currently approved for use in the EU (Fig. 2). The decision to cease using these two substances at EU level came into effect after data were collected for the most recent usage reports in Ireland. It will be several years before the effects of EU regulation on AS use can be discerned in Irish national PPP usage data due to the legally-required five year reporting frequencies, and this impedes the short-term evaluation of the effectiveness of EU pesticide regulation.

The countries with the largest national cultivated areas (CA), and therefore the largest potential area of PPP usage, were Spain and Germany. Spanish CA was 24 times the CA of Estonia, 10 times that of Finland, five times that of Ireland and 1.4 times that of Germany. However, the country with the greatest minimum application areas of fungicides and plant growth regulators was Germany, where prothioconazole is applied to 3,383 thousand ha and trinexapac was applied to 3,033 thousand ha (Table 5). The minimum potential area of herbicide application was similar in Spain and Germany, but the minimum potential area of insecticide use was slightly greater in Spain (dimethoate applied to 826,432 ha), than in Germany (thiacloprid was applied to 812,624 ha). With the exception of plant growth regulators, the minimum area of application for all groups of PPP as a proportion of the total CA was smallest in Ireland (Fig. 3; right panels). Based on minimum potential areas, both herbicide (56% CA) and insecticide (6% CA) were most widely applied in Finland, and fungicides are applied to a larger proportion of CA in Germany (20%) than in any other country. As a general trend for all the countries assessed, the minimum potential area of insecticide application was smaller than that for herbicides or fungicides (from 1.6 to 8.8 times smaller) and represents less than 10% of CA. The minimum potential application of plant growth regulators as a proportion of total cultivated area was highly variable, accounting for the smallest proportional area of all PPP groups in Spain (0.2%), and the second greatest proportional area in Germany (18%). Overall, differences among countries are probably related to distinct farming systems, crop preferences, agronomic culture and traditions, and/or climatic conditions making areas more or less susceptible to pest damage.

Figure 3 Minimum and maximum areas of PPP active substances applied for each group and country.

Minimum (black) and maximum potential areas (grey) where PPP active substances (ASs) were applied in thousands of hectares and as a percentage of the total cultivated area per country and per PPP group. For all countries except for Finland, the minimum area is equal to the basic area of the most widely applied AI for each PPP group because it was not possible to determine the degree of overlap PPP application for the different active ingredients belonging to the same PPP group. In the case of the Finish database, basic areas provided are disaggregated by PPP group but not by active ingredients. The maximum potential area represents the entire national cultivated areas for all countries. Data sources: public accessible data of national plant protection products use, and EUROSTAT TAG00025 databases.

Where basic areas were reported for AS use, as was the case for the Member States included in our analysis (except for Finland), a comparison of AS use in different countries could be made. However, this approach does not allow for an estimate of the total area within a country treated with broader PPP groups, as different ASs can be applied to the same cropped area either individually or in combination. For instance, in the Irish dataset for arable fields, a maximum and average number of 29 and 12 different ASs were used per field, respectively. It is therefore very difficult to compare overall PPP usage at a broad level across different countries because they are reported at AS level with no information on the degree of overlap in the area of application. Here we chose to use the most widely applied ASs per PPP group as an estimate of the minimum value of total basic area per PPP group to try to alleviate this issue. However, the risk of underestimation is high, particularly for those crops where a combination of ASs is commonly applied. In fact, a comparison of results based on broader PPP group- and AS-disaggregated basic areas (Figs. 3 and 1, respectively), reveals that the basic areas for PPP groups are from 1.05 (for plant growth regulators) to 2.71 (for herbicides) times higher than the estimate of minimum area affected based on AS usage data.

We chose to use basic area as the unit of measurement to compare pesticide use in this study, as understanding the geographic extent of agricultural land treated with particular pesticides is a first step in considering the risk of pesticides to the environment. However, a second step would be to include the intensity of use including the number of applications to a particular cropped area, which is often summarised as spray hectares. Some cropped areas receive numerous applications of the same substances, which could dramatically influence the risks particular pesticides have to the environment. For example, many grassland systems may only receive single or low numbers of applications, whereas cereals can have many more applications and fruit and vegetables can have even more, with an average of 17 applications per cropped area recorded in the UK (Goulson, Thompson & Croombs, 2018; van Drooge, Groeneveld & Schipper, 2001). Taking measures of intensity into account could reveal different patterns in the relative amounts of different pesticide groups used to what is reported here for basic area, and should be considered for further study and full scale risk assessment.

The data used in our study (i.e., basic area of PPP use) are broadly consistent with available PPP sales data available (EUROSTAT, 2019a, EUROSTAT, 2019b) for the years assessed. Annual PPP sales (in kg) were the highest in Germany and Spain, both of which are the countries with the largest basic area treated with PPPs. Interestingly, sales of fungicides in Spain were much higher than that of Germany, even though our analysis showed the estimated basic area of fungicide use in Spain was lower. This may indicate that fungicides are applied more often to the same parcel of land in Spain than in Germany (i.e., there is a higher intensity of use). It may also indicate that a greater variety of substances are being used (and therefore not captured in our minimum estimate which was based on the most widely used substance), or that the substances being used are applied in greater quantities.

Another relevant issue is the suitability of including other sectors apart from agriculture, such as forestry or amenity use, in the PPP usage data reported by EU Member States. For example, forestry represents a significant land-use in some European countries in terms of PPP use. This is the case in Ireland, where forestry covers over 770,000 ha, an area greater than that of arable land, and where both herbicide and insecticide application occurs. However, data on these other sectors are not currently summarized or collected with the agricultural data within the EU, which makes the estimation of national or European total PPP use challenging and hampers the development of holistic risk assessments at a landscape level that needs the integration of spatial-explicit PPP use information with other data sets (e.g., soil type maps) at national scale.

Conclusions

Recommendations to improve national statistics of PPP use in EU

According to Schulz et al. (2021), the unavailability of open-access pesticide use data occurs globally, which hampers the application of advanced risk assessment approaches developed by the scientific community (e.g., Sponsler et al., 2019) to evaluate one of the crucial drivers of global biodiversity decline. Our study demonstrates the current gaps in data collection and reporting on national PPP use across the EU. Unlike sales data for EU Member States, PPP usage data are not publicly available for all countries. When available, these data are presented in heterogeneous format, different units, and the crop classifications and PPP groups are not consistent. We have tried to summarise the basic area of PPP application across European countries as a first crucial step to upscale potential risks to the environment to national and international scales. While we have shown that it is possible for different countries to record comparable PPP usage data, the current regulations governing pesticide usage data reporting do not result in consistent and accessible data at EU scale.

The recently published EU “Farm to fork strategy for a fair, healthy and environmentally-friendly food system” states that the European Commission will also propose changes to the 2009 regulation concerning statistics on PPPs to overcome data gaps and promote evidence-based policymaking. This would be an important step in ensuring the usability of these data for multiple purposes, and we echo other calls to improve this process (e.g., Mesnage et al., 2021).

Based on our results, our main recommendations are:

(a) Consistent reporting requirements around data format, disaggregation levels of PPP groups and crop types and reporting years.

(b) Mandatory reporting of basic and treated areas. Basic areas tell us about the geographical extent of PPP and respective AS use, while treated areas give us information about how frequently these PPPs and ASs are applied in a given crop type or country. Both metrics are useful as the information they carry is complementary.

(c) Mandatory reporting of summary data.

(d) Sub-national georeferenced data of pesticide usage should be provided if detailed environmental risk assessment is a priority.

(e) Controlled vocabularies (e.g., same PPP groups for all countries) to assure harmonisation among national databases. Additionally, in the case of crop types, these should be named in accordance with the cultivation types shown in EUROSTAT database (Table 3).

(f) Consideration of all sectors that use PPPs, including sectors such as forestry and amenity use.

(g) Accessibility and usability of all these data should be guaranteed since current data repository platforms are already available.

The EU has one of the most developed systems for collection of national pesticide usage statistics globally. Increasing the uniformity and accessibility of pesticide usage data in the EU would further enhance their usefulness to design and evaluate policy actions that ensure sustainable agricultural practices that promote both food production and a healthy environment.

We thank Stephen Jess, Tom Medlycott, and the PROTECTS project team for useful discussions.

Additional Information and Declarations

Competing Interests

Author Contributions

Data Availability

James Quirke is an employee of the Irish Department of Agriculture, Food and the Marine.

Ana López-Ballesteros conceived and designed the experiments, performed the experiments, analyzed the data, prepared figures and/or tables, authored or reviewed drafts of the article, and approved the final draft.

Aoife Delaney conceived and designed the experiments, performed the experiments, analyzed the data, prepared figures and/or tables, authored or reviewed drafts of the article, and approved the final draft.

James Quirke conceived and designed the experiments, authored or reviewed drafts of the article, and approved the final draft.

Jane C. Stout conceived and designed the experiments, authored or reviewed drafts of the article, and approved the final draft.

Matthew Saunders conceived and designed the experiments, authored or reviewed drafts of the article, and approved the final draft.

James C. Carolan conceived and designed the experiments, authored or reviewed drafts of the article, and approved the final draft.

Blánaid White conceived and designed the experiments, authored or reviewed drafts of the article, and approved the final draft.

Dara A. Stanley conceived and designed the experiments, performed the experiments, authored or reviewed drafts of the article, and approved the final draft.

The following information was supplied regarding data availability:

All the data is available from pesticide usage data repositories:

- Estonia: https://andmed.stat.ee/en/stat: search for Environment > Agri-environmental-indicators > KK2081: QUANTITY OF PESTICIDES USED AND THE BASIC AREA TREATED IN AGRICULTURAL HOLDINGS BY ACTIVE SUBSTANCE AND CROP. Data selected for all crops and active substances in 2015 and the indicator of “basic area treated (ha)”.

- Finland: https://statdb.luke.fi/PXWeb/pxweb/en/LUKE/LUKE__02%20Maatalous__04%20Tuotanto__34%20Kasvinsuojeluaineiden%20kaytto%20maataloudessa/02_Kasvinsuojeluainekaytto.px/: Select 2018, all crops, all targets, and in regards to pesticides, the variables: “area under cultivation, 1,000 ha” and “treated area, % of area under cultivation” (which is basic area).

- Ireland: http://www.pcs.agriculture.gov.ie/sud/pesticidestatistics/

- Germany: https://papa.julius-kuehn.de/index.php?menuid=33

- Spain: https://www.mapa.gob.es/es/estadistica/temas/estadisticas-agrarias/agricultura/estadisticas-medios-produccion/fitosanitarios.aspx.

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
