# Peer review of "Assessing availability of European plant protection product data: an example evaluating basic area treated"

_PeerJ, doi:10.7717/peerj.13586_

## Round 0.1 · original submission · Major Revisions

Your manuscript needs major revisions. That are many good comments from both reviewers for you to consider and respond as appropriate. One overarching comment you might consider is related to the title, which possibly overstates the scope of the paper.

Reviewer 1 ·

Basic reporting

Ok

Experimental design

There's no experiment

Validity of the findings

See below my comments

Additional comments

The authors estimated the area applied with PPPs in Ireland and identified the most used AIs. The paper reads more like a review rather than a research paper.
Although a national-based estimate is a good first step, assessing contamination level and exposure of non-target organisms require a much finer scale data than national-level data. The dynamic, fate, transport, and residues of pesticides, like any other chemicals, depend largely on soil properties and climatic conditions. Also spatially distributed data is important for understanding the potential exposure of non-targeted organisms to the AIs. If the pesticide were applied in places where biodiversity is low, then the risk for non-target organisms getting in-contact with the pesticide can also be low. Without spatial data, it is difficult to do a meaningful risk assessment. The authors should consider compiling subnational pesticide usage data.
Line 378-409: The risk assessment was not correctly conducted. First of all, the LC50 is expressed as concentrations. How did you convert the application rate into concentrations in water, soil, and atmosphere? How did you decide the High, low and medium risk? Some AIs are not soluble or have high adsorption to soil, thus, these chemical are very unlikely to get to water even though they might be very toxic to fish. Similarly, some AIs have very low volatilization, thus may not even get to the atmosphere. Following my comment above, risk assessment is only meaning when it is done with spatially distributed data and by considering the physiochemical properties of the AIs, the soil properties, and the environmental conditions (temperature, rainfall, groundwater, proximity to surface water, etc). I recommend removing the risk assessment part from this manuscript and also removing it from the title. Otherwise, the risk assessment has to be done properly.
Other comments:
Line 47: FAOSTAT records about 4 million tonnes of pesticides usage globally in 2015. I think it should be 3 billion kgs (not tonnes) in 2009. Please check and update the figure. FAOSTAT provides data up to 2019.
Line 91 – 114: Besides FAOSTAT and EU commission, USGS reports usage of individual AI in different cropping systems at county level. PESTCHEMGRID also estimates the individual AI usage globally (including Ireland) in different cropping systems at georeferenced grid level. I think these databases should be mentioned.
Line 197: “…. grasslands alone account for 99.4% of this crop type” – Which crop type?
Line 199-200: Is the fraction of land applied with PPPs coincide with the fraction of organic farming?
Line 206: “These represent 44%, 32% and 24% of the area where herbicides are applied.” I do not understand this sentence. Do you mean PPPs rather than herbicides?
Line 209 – 212: How do these results compared to FAOSTAT in terms of unit total mass and how do they compare against PESTCHEMGRIDS in terms of both mass and surface area?

·

Basic reporting

The english is clear and the paper reads well. Literature referenced is appropriate, although balance in relation to a summary of the regulatory system would be helpful. The title possibly overstates the scope of the paper, the main output of which is the assessment of data availability and recommendations for improvement to support better interpretation of comparability and impact.

Experimental design

The comparison of usage data between European MS, and recommendations for improvement of recording, to allow comparison of impacts is relevant and meaningful. The recommended proxy of basic treated area I don't think is adequately demonstrated to be an appropriate comparator nor are the associated maximum and minimum treated areas. Additional comments have been added to the annotated PDF. The comparison of commonly used PPPs against regulatory ecotoxicological endpoints isn't novel and basing this comparison on basic areas of top 5 PPPs doesn't adequately demonstrate an advantage to other methods of comparison such as use of pesticide impact indices using PPP weight. The methods are not quite described with sufficient detail to fully understand or replicate.

Validity of the findings

The analysis of the available data fully support the recommendations for the need for consistent data collection and reporting and this is an important and valuable message. The recommendation on the suitability of basic area to be a suitable proxy for use is not adequately supported in my opinion, either for use as a comparison of usage between MS or to indicate comparative implications for impact on non-target organisms.

---

## Round 0.2 · accepted · Accept

Thank you for your efforts in revising your manuscript according to reviewer comments.

Reviewer 1 ·

Basic reporting

No comment

Experimental design

No comment

Validity of the findings

No comment